# Optimal Alternative for Quantifying Reference Evapotranspiration in Northern Xinjiang



**Ping Jiao [1,2,3] and Shun-Jun Hu [1,3,*]**

1   State Key Laboratory of Desert and Oasis Ecology, Xinjiang Institute of Ecology and Geography, Chinese Academy of Sciences, Urumqi 830011, China; jiaoping20@mails.ucas.ac.cn
2   School of Resources and Environment, University of Chinese Academy of Sciences, Beijing 100049, China
3   National Field Scientific Observation and Research Station of Akesu Oasis Farmland Ecosystem, Aksu 843017, China
*   Correspondence: xjhushunjun@aliyun.com; Tel.: +86-135-7997-6452

**Abstract:** Accurate estimation of reference evapotranspiration is a key step in irrigation and water resources planning. The Penman Monteith (FAO56-PM) formula recommended by FAO56-PM is the standard for calculating the reference evapotranspiration. However, the FAO56-PM model is limited in the observation of meteorological variables, so it is necessary to choose an alternative $ET_0$ model which requires less meteorological data. Based on the daily climate data of eight meteorological stations in northern Xinjiang from 2000 to 2020, seven empirical models (Hargreaves, Berti, Dorji, Dalton, Meyer, WMO, Albrecht) and four optimization algorithms (RF model, LS-SVR model, Bi-LSTM model and GA-BP model) combined with seven different parameters were evaluated comprehensively. The results show that the accurate of the empirical model based on temperature is obviously better than the empirical model based on air mass transport. The annual and multi-year alternative $ET_0$ models of different input parameter combinations are: LS-SVR1, RF2, LS-SVR3, LS-SVR4, GA-BP5, LS-SVR6, GA-BP7. It can be used as a substitute for the reference evapotranspiration model without relevant meteorological data. Only the LS-SVR6 model and GA-BP7 model are recommended as the best alternative models for northern Xinjiang reference evapotranspiration at daily, monthly and seasonal scales.

**Keywords:** reference evapotranspiration; empirical model; regression prediction algorithms; optimal alternative

## 1. Introduction

The quantification and accurate estimation of evapotranspiration are of great significance to the formulation of farmland irrigation systems, the study of hydrology and water balance, and the planning of water resources. The Penman Monteith (FAO56-PM) method recommended by the FAO [1] is the standard method for estimating reference evapotranspiration ($ET_0$) thus far [2,3]. This method does not need to be initially calibrated locally and has global applicability [4–6]. However, this method has many input parameters (air temperature, humidity, solar radiation and wind speed), so the acquisition of many meteorological parameters becomes the only limitation of its application [7]. In this case, relatively simple empirical models are usually used to estimate $ET_0$, and the selection of the optimal empirical model is of great significance to water resource planning and management [8,9]. Existing empirical evapotranspiration models mainly include temperature-based, radiation-based, air-mass-transport-based and combination models. The results show that the combined models are better, followed by radiation-based, temperature-based and aerodynamic models [10]. However, the combined model is highly dependent on the input of meteorological variables, which requires the sufficient input of meteorological variables, such as net radiation, soil heat flux, air temperature, wind speed and relative humidity. Moreover, some scholars also pointed out that, compared

with the climate in humid and semi-humid regions, $ET_0$ in arid and semi-arid regions is mainly affected by aerodynamics and water vapor pressure deficit rather than radiation availability [11,12]. Therefore, the $ET_0$ model based on temperature and air mass transport was selected in this study to verify its applicability in northern Xinjiang.

In recent years, more scholars have applied artificial intelligence (AI) to the prediction and estimation of $ET_0$ and published a number of papers. For example, Vahid Nourani et al. [13] used a feed-forward neural network (FFNN), adaptive neuro-fuzzy reasoning system (ANFIS), support vector machine regression (SVR) and other algorithms to simulate the $ET_0$ several climate zones of Turkey, Cyprus, Iraq, Iran and Libya. Ahmed Elbeltagi et al. [14] estimated long-term $ET_0$ in Egypt through a deep neural network (DNN). Babak Mohammadi and Saeid Mehdizadeh [15] used support vector regression (SVR) and random forest (RF) to simulate daily $ET_0$ from Isfahan, Urmiya and Yazd, Iran. Tongren Xu et al. [16] evaluated the adaptability of machine learning, remote sensing and land surface $ET_0$ products in the United States. Saman Maroufpoor et al. [17] simulated $ET_0$ in Iran by using artificial neural network optimization (ANN-GWO). Francesco Granata [18] used the M5P regression tree, bagging, random forest (RF) and support vector machine regression (SVR) to simulate $ET_0$ in central Florida. Juan Yin et al. [19] simulated $ET_0$ in the central Ningxia region of China by using the mixed bidirectional long- and short-term memory model (BI-LSTM). Jingran Liu et al. [20] predicted the actual evapotranspiration of green pepper by using an extended neural network (MEA-ENN, GA-ENN) optimized by the mind evolutionary algorithm and genetic algorithm. The results show that artificial intelligence can accurately predict daily $ET_0$ and is a powerful tool for modelling $ET_0$ using incomplete meteorological parameters.

In summary, this study selected eight widely used empirical models for performance evaluation, including four temperature-based models: Hargreaves model [21], Berti model [22] and Dorji model [23] and four mass transfer-based models: Dalton Model [24], Meyer Model [25], WMO Model [26] and Albrecht Model [27]. Four optimization algorithms (RF model, LS-SVR model, BI-LSTM model and GA-BP model) were combined with 28 optimization algorithm models to make a comprehensive ranking. On this basis, this study proposes the following hypotheses. First, the empirical model and the $ET_0$ model based on the optimization algorithm are significantly affected by time and space in northern Xinjiang. Second, simple linear regression and the global performance index GPI can effectively verify the performance effect of the $ET_0$ model in northern Xinjiang. The research objectives of this study are: (1) to use linear regression to verify seven empirical models to determine the best substitute for the FAO56-PM model in northern Xinjiang; (2) to use the global performance index GPI to rank the 28 models based on the optimization algorithm and to determine the best algorithm model under seven different parameter input combinations; and (3) to discuss the influence of time scale on the model and to recommend the most appropriate reference evapotranspiration estimation model for northern Xinjiang at different time scales. The innovation of this study lies in the first application of eight empirical models and four models based on an optimization algorithm in northern Xinjiang and the recommendation of corresponding models in the absence of meteorological data to fill the gap in knowledge regarding models suitable for northern Xinjiang under different meteorological parameters.

## 2. Overview of the Study Area and Data

### 2.1. Study Area

Xinjiang is located in the hinterland of Eurasia and the northwest border of China. The Tianshan Mountains traverse the whole territory, dividing the whole territory into northern and southern Xinjiang. The part north of the Tianshan Mountains is called northern Xinjiang. In northern Xinjiang lies the Altai Mountains, and between the Altai Mountains and Tianshan Mountains lies the semi-closed Junggar Basin (Figure 1). Northern Xinjiang has a temperate continental arid climate, and the annual average temperature ranges from 1.96 °C to 9.06 °C. The annual precipitation ranges from 137.75 mm to 253.96 mm, and

the annual evaporation ranges from 909.92 mm to 1328.76 mm (Table 1). In the eastern and western regions, the wind speed is high, and the number of windy days is high. The mountainous terrain is undulating, and the wind is blocked. The number of gale days in the plains area is greater than that in the middle and low mountainous areas, and the number of gale days in the Alashankou area in the western Junggar Basin is the highest. In short, due to the dry climate, less precipitation, sparse vegetation, loose soil quality and high winds, the annual $ET_0$ in northern Xinjiang is relatively large, and it is vulnerable to drought stress.

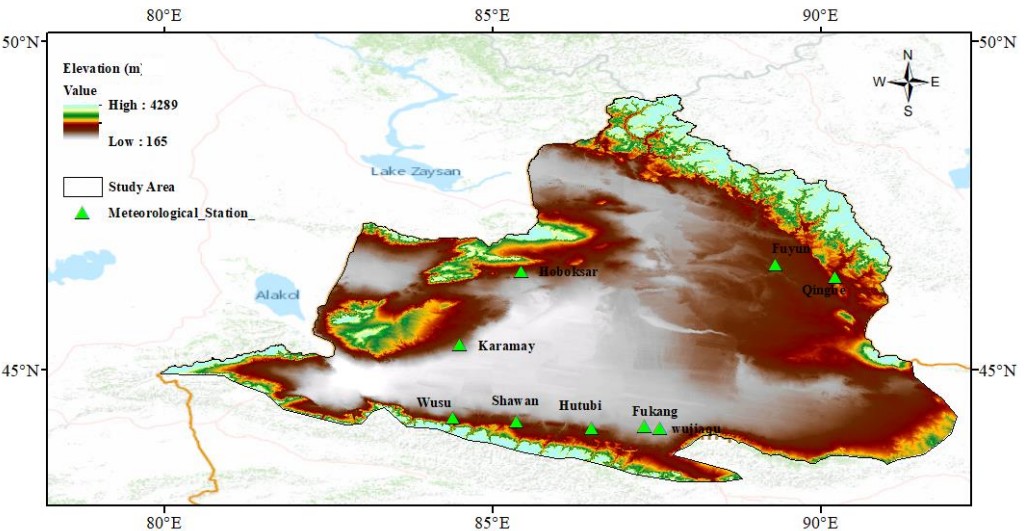

**Figure 1.** Geographical locations of Gurbantunggut Desert and meteorological stations.

**Table 1.** Geographical and long-term average meteorological information for 8 stations in Gurbantunggut Desert, Xingjiang, China. The values in parentheses are the standard deviations for each variable.

| Station | Lon (°E) | Lat (°S) | DEM (m) | T (°C) | RH (%) | U (m/s) | VPD (kPa) | Rainfall (mm) | $ET_0$ (mm) | AI |
|---------|----------|----------|---------|--------|--------|---------|-----------|---------------|-------------|-----|
| Shawan | 85.37 | 44.2 | 522.2 | 7.94 (15.22) | 64.14 (18.35) | 1.17 (0.65) | 0.96 (0.90) | 253.96 (2.29) | 1086.02 (2.34) | 4.28 |
| Wujiaqu | 87.32 | 44.12 | 440.5 | 7.10 (16.49) | 59.94 (19.41) | 1.31 (0.69) | 1.13 (1.07) | 169.39 (1.70) | 1155.58 (2.50) | 6.82 |
| Fuyun | 89.31 | 46.59 | 807.5 | 4.53 (15.89) | 57.15 (19.05) | 1.34 (1.07) | 0.93 (0.89) | 230.12 (2.24) | 1028.77 (2.42) | 4.47 |
| Hoboksar | 85.45 | 46.49 | 1322.1 | 4.45 (12.50) | 53.71 (15.78) | 1.89 (1.76) | 0.75 (0.64) | 174.92 (1.87) | 1038.12 (2.18) | 5.93 |
| Qinghe | 90.23 | 46.4 | 1218.2 | 1.96 (15.63) | 57.81 (16.31) | 1.01 (0.68) | 0.75 (0.69) | 214.06 (2.19) | 909.92 (2.03) | 4.25 |
| Karamay | 84.51 | 45.37 | 450.3 | 9.06 (16.00) | 50.28 (21.78) | 1.85 (1.18) | 1.25 (1.17) | 137.75 (1.56) | 1328.76 (2.89) | 9.65 |
| Wusu | 84.4 | 44.26 | 478.7 | 8.92 (15.31) | 57.59 (20.12) | 1.25 (0.59) | 1.11 (1.06) | 214.14 (2.14) | 1106.45 (2.38) | 5.17 |
| Hutubi | 86.51 | 44.1 | 575.1 | 8.15 (15.64) | 59.48 (20.86) | 1.59 (0.76) | 0.37 (1.04) | 210.05 (1.92) | 1192.17 (2.62) | 5.68 |

## 2.2. Data Sources

Eight weather stations in northern Xinjiang were selected in this study (Figure 1), and daily meteorological data were collected from 2000 to 2020. The observation data include temperature, humidity, wind speed, rainfall, saturated water vapour pressure deficit and extraterrestrial radiation (Table 1). The climatic data of historical days are obtained from the China Meteorological Data Sharing Service System (http://data.cma.cn (access on

 Among them, the Wujiaqu, Wusu, Shawan and Qinghe datasets span from 2000 to 2017, and the datasets of Hutubi, Fuyun, Hebukesel and Karamay span from 2000 to 2020.

## 3. Modelling Structure and Approach

### 3.1. FAO56 Penman–Monteith Model

The FAO56 P-M approach proposed by Allen et al. [1] served here as a reference method, it is expressed by Equation (1):

$$ET_0 = \frac{0.408\Delta(R_n - G) + \gamma\frac{900}{T + 273}U_2(e_s - e_a)}{\Delta + \gamma(1 + 0.34U_2)} \tag{1}$$

where $ET_0$ is the reference evapotranspiration (mm day$^{-1}$), $Rn$ is net radiation (MJm$^{-2}$ day$^{-1}$), $G$ is soil heat flux (MJm$^{-2}$ day$^{-1}$), $\gamma$ is the psychometric constant (kPa °C$^{-1}$), $T$ is the mean air temperature (°C), $U_2$ is the wind speed at 2 m height (ms$^{-1}$), $e_s$ is the saturation vapour pressure (kPa), $e_a$ is the actual vapour pressure (kPa), and $\Delta$ is the slope of the vapour pressure curve (kPa °C$^{-1}$).

### 3.2. Empirical Models

Temperature-based models performed well in all subregions of the world. Due to the special climatic conditions in desert areas, it is necessary to select an empirical model that is more suitable for $ET_0$ in northern Xinjiang through model selection. Hargreaves and Allen, Berti and Dorji models were selected for comparison with FAO56 P-M models (Table 2). Similarly, four empirical models based on air mass transport, Dalton, Meyer, WMO and Albrecht, were selected to verify the applicability of the four models in northern Xinjiang to find an alternative model with higher applicability under fewer input meteorological parameters (Table 2).

**Table 2.** Empirical reference evapotranspiration model and parameter calculation.

| Meteorological Inputs | Equations | Proposed by |
|---|---|---|
| Based-temperature $ET_0$ models | | |
| $R_a$, $T_{max}$, $T_{min}$, | $ET_0 = 0.0023R_a(T_{max} - T_{min})^{0.5}(T + 17.8)$ | Hargreaves and Allen [21] |
| $R_a$, $T_{max}$, $T_{min}$, | $ET_0 = 0.408 \times 0.00193R_a(T_{max} - T_{min})^{0.517}(T + 17.8)$ | Berti [22] |
| $R_a$, $T_{max}$, $T_{min}$, | $ET_0 = 0.408 \times 0.002R_a(T_{max} - T_{min})^{0.293}(T + 33.9)$ | Dorji [23] |
| Mass transfer-based $ET_0$ models | | |
| $U_2$, $e_s$, $e_a$ | $ET_0 = (0.3648 + 0.7223U_2)(e_s - e_a)$ | Dalton [24] |
| $U_2$, $e_s$, $e_a$ | $ET_0 = (0.375 + 0.05026U_2)(e_s - e_a)$ | Meyer [25] |
| $U_2$, $e_s$, $e_a$ | $ET_0 = (0.1298 + 0.0934U_2)(e_s - e_a)$ | WMO [26] |
| $U_2$, $e_s$, $e_a$ | $ET_0 = (0.1005 + 0.297U_2)(e_s - e_a)$ | Albrecht [27] |

$Ra$ is the theoretical radiation, $T_{max}$ is the daily maximum temperature, $T_{min}$ is the daily minimum temperature, and other parameters are the same as above.

### 3.3. Random Forest-Based Reference Evapotranspiration (ET₀) Model

Random forest (RF) is a class of discriminant models that support classification, regression, and multi-classification [28]. It is based on bagging integration on decision trees and introduces random attribute selection of prediction in the training process of decision trees. It establishes a forest in a random way. The forest is composed of many decision trees. There is no correlation between each decision tree, and with the same distribution for all trees, multiple trees are used to train and predict the samples [29].

For a given classifier set, the numerical estimator is $h_1(X)$, $h_2(X)$, ... , $h_k(X)$, and a random vector $\Theta$ is set; therefore, the tree predictor $h(X, \Theta)$ can take on numerical values.

As the number of trees in the forest increases, the training set drawn at random from the distribution of the random vector $Y$, $X$, and the marginal functions can be given as:

$$mg(X,Y) = av_k I(h_k(X) = Y) - \max_{j \neq Y} av_k I(h_k(X) = j) \tag{2}$$

For a large number of decision trees, the following two basic theorems are valid:

**Theorem 1.** *We define the upper bound for generalization error of RF as follows:*

$$PE^* = P_{X,Y}(mg(X,Y) < 0) \tag{3}$$

**Theorem 2.** *As the number of decision trees increase, all sequences will finally converge to:*

$$P_{X,Y}(P_\Theta h(X,\Theta) = Y) - \max_{j \neq 1} P_\Theta(h(X,\Theta) = j) < 0) \tag{4}$$

*3.4. Least Square Support Vector Regression*

A support vector machine (SVM) was introduced in the 1990s [30] and then extended for support vector regression (SVR), which is a new machine learning algorithm based on the structural risk minimization criterion. The biggest problem of traditional neural networks is that they can minimize the training error. However, it cannot minimize the generalization error in the learning process. SVM is successfully applied for the classification and prediction based on VC dimensions and structural risk minimization. Then, the least square support vector machine (LS-SVR) was proposed by Suykens and Vandewalle [31] of the United States on this basis. The difference between LS-SVR and SVM is that the relaxation variables in the optimization objective are changed from a penalty term to a quadratic term, and the constraint conditions are also changed into only equality constraints, which allows the solution to be realized by the least square method.

The linear regression process is as follows: the n-dimensional inputted vector and 1-dimensional outputted vector: $(x_k, y_k)$, $x_k \in R^n$, $y_k \in R$, $k = 1, \ldots, N$, are mapped from the original space to the high-dimensional space $F$, and the optimal linear regression function is constructed in $F$. That is:

$$y(x) = \omega \varphi(x) + b, \ \ x_k \in R^n, \ b \in R \tag{5}$$

According to the principle of structural risk minimization, there are:

$$J(\omega, R_{emp}) = \frac{1}{2}\|\omega\|^2 + \frac{1}{2}\gamma R_{emp} \tag{6}$$

where, $\|\omega\|^2$ is the complexity of control model, $\gamma$ is the determining penalty factor, $R_{emp}$ is the loss function, LS-SVM selects the two-norm of e as the loss function. At this time, the optimization problem becomes:

$$\min_{\omega,b,\xi} J(\omega, e) = \frac{1}{2}\omega^T \omega + \frac{1}{2}\gamma \sum_{k=1}^{N} e_k^2 \tag{7}$$

The constraint condition is:

$$y_k = \omega^T \varphi(x_k) + b + e_k, \ \ k = 1, \ldots, N \tag{8}$$

The model is changed into dual space and Lagrange function is introduced:

$$L(\omega, b, e, a) = \frac{1}{2}\omega^T \omega + \frac{1}{2}\gamma \sum_{k=1}^{N} e_k^2 - \sum_{k=1}^{N} \alpha_k \left\{ \omega^T \varphi(x_k) + b - y_k + e_k \right\} \tag{9}$$

The partial derivation of the above formula is obtained:

$$
\begin{cases}
\omega = \sum\limits_{k=1}^{N} \alpha_k \varphi(x_k) \\[2mm]
\sum\limits_{k=1}^{N} \alpha_k = 0 \\[2mm]
\alpha_k = \frac{1}{2}\gamma e_k \\[2mm]
\omega^T \varphi(x_k) + b - y_k + e_k = 0
\end{cases}
\tag{10}
$$

For $k = 1, \ldots, N$, the system of linear equations is obtained by eliminating $\omega$ and $e$:

$$
\left\{
\begin{array}{cc}
0 & 1_v^T \\
1_v & \Omega + \gamma^{-1}I
\end{array}
\right\}
\left[
\begin{array}{c}
b \\
a
\end{array}
\right]
=
\left[
\begin{array}{c}
0 \\
y
\end{array}
\right]
\tag{11}
$$

Obtained:

$$
\begin{cases}
y = [y_1, \cdots, y_N]^T \\[1mm]
\alpha = [\alpha_1, \cdots, \alpha_N]^T \\[1mm]
1_v = [1, \cdots, 1]^T \\[1mm]
\Omega_{ki} = \varphi^T(x_k)\varphi(x_i)
\end{cases}
\quad k, i = 1, \ldots, N
\tag{12}
$$

where, $I$ is the identity matrix.

$a$ and $b$ are calculated by the least square method:

$$
y(x) = \sum_{k=1}^{N} \alpha_k \varphi^T(x)\varphi(x_k) + b
\tag{13}
$$

Let $K(x, x_k) = \varphi^T(x)\varphi(x_k)$, the prediction model of LS-SVM is as follows

$$
y(x) = \sum_{k=1}^{N} \alpha_k K(x, x_k) + b, \quad k = 1, \ldots, N
\tag{14}
$$

where $K(x, x_k)$ is the kernel function and the kernel function realizes the mapping from low-dimensional space to high-dimensional space and transforms the non-linear problem of low-dimensional space into a linear problem of high-dimensional space.

### 3.5. Bidirectional Long-Term and Short-Term Memory Network

The bidirectional long-term and short-term memory network (LSTM) cell, first proposed by Hochreiter and Schmidhuber [32], is an upgraded version of a recursive neural network that can overcome the problem of vanishing recurrent neural network (RNN) gradients. Standard LSTM networks deal with sequences chronologically, and they ignore future contexts. To overcome this shortcoming, the Bi-LSTM algorithm is introduced to improve the accuracy. The algorithm is a deformed structure of LSTM, and the forward and backward LSTM layers contained in the memory block are able to utilize past and future information. The hidden layer with opposite time series is obtained, and the two hidden layers are connected to obtain the same output result. The hidden layer of Bi-LSTM at time t includes forward $\overrightarrow{h_t}$ and backward $\overleftarrow{h_t}$ bars:

$$
\overrightarrow{h_t} = \overrightarrow{LSTM}(h_{t-1}, x_t, c_{t-1}), \ t \in [1, \ T]
\tag{15}
$$

$$
\overleftarrow{h_t} = \overleftarrow{LSTM}(h_{t+1}, x_t, c_{t+1}), \ t \in [T, \ 1]
\tag{16}
$$

$$H_t = \left[\overrightarrow{h_t}, \overleftarrow{h_t}\right] \tag{17}$$

where $T$ denotes the length of the time.

### 3.6. Back Propagation Neural Network Optimized by Genetic Algorithm

The genetic algorithm (GA) is a heuristic algorithm simulating biological evolution that has three genetic operators: selection, crossover and mutation [33]. BPNN consists of a three-layer network structure of an input layer, hidden layer and output layer [12]. With neurons as the basic unit, the BPNN solves the non-linear fitting problem mainly by setting activation functions in the hidden layer and output layer. In this study, a genetic algorithm was used to optimize the structural parameters of the BP neural network. To realize the dynamic updating of the value interval of the number of gene loci in the hidden layer, the crossover operator and mutation operator of the structural parameter chromosome in GA-BP were improved. First, the neuron gene calculation was carried out, and then the value interval of gene loci was derived according to the number of neurons to generate the value of gene loci. The BP structural parameter population was optimized by a genetic operator to obtain the optimal structural parameters (Figure 2).

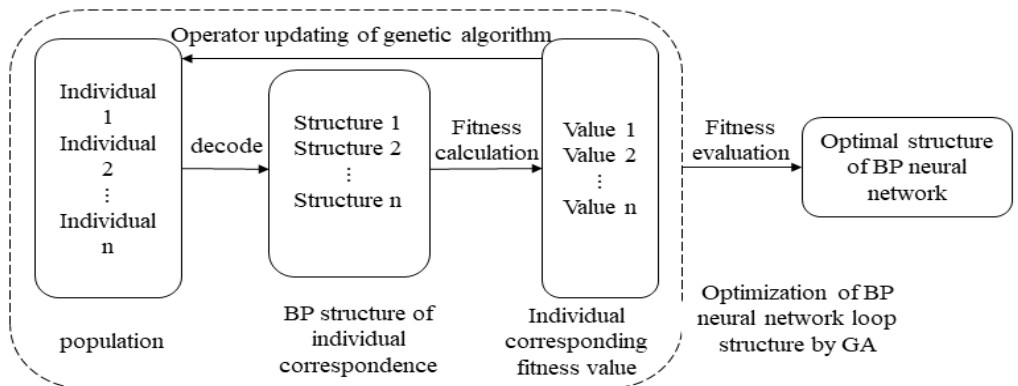

**Figure 2.** Optimization of BP neural network structure parameters by GA.

Through the correlation analysis of meteorological factors (Figure 3), it can be seen that under the condition of $p < 0.05$, all parameters are significantly correlated with $ET_0$.

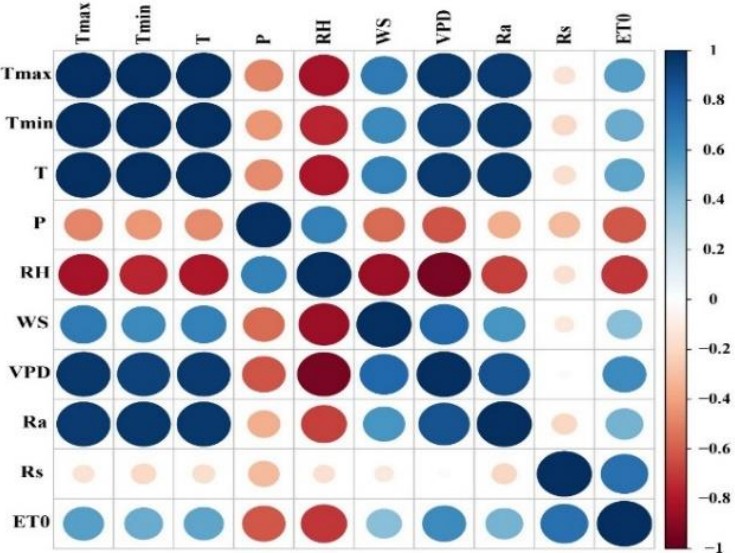

**Figure 3.** Correlation analysis of meteorological factors.

Therefore, the following parameters (Table 3) are selected for $ET_0$ prediction. We adopted four optimization algorithms, including RF, LS-SVR, BI-LSTM and GA-BP, and combined with different parameter combinations; a total of 28 reference evapotranspiration models were established.

**Table 3.** Summary of the inputs used for the implementation of each RF, LS-SVR, Bi-LSTM and GA-BP models.

| Algorithm | | | | $T$ | $T_{max}$ | $T_{min}$ | $Ra$ | $Rs$ | $RH$ | $U_2$ |
|---|---|---|---|---|---|---|---|---|---|---|
| RF1 | LS-SVR1 | Bi-LSTM1 | GA-BP1 | ✔ | ✔ | ✔ | ✔ | | | |
| RF2 | LS-SVR2 | Bi-LSTM2 | GA-BP2 | ✔ | ✔ | ✔ | ✔ | ✔ | | |
| RF3 | LS-SVR3 | Bi-LSTM3 | GA-BP3 | ✔ | ✔ | ✔ | ✔ | | ✔ | |
| RF4 | LS-SVR4 | Bi-LSTM4 | GA-BP4 | ✔ | ✔ | ✔ | ✔ | ✔ | ✔ | |
| RF5 | LS-SVR5 | Bi-LSTM5 | GA-BP5 | ✔ | ✔ | ✔ | ✔ | | ✔ | ✔ |
| RF6 | LS-SVR6 | Bi-LSTM6 | GA-BP6 | ✔ | ✔ | ✔ | ✔ | ✔ | | ✔ |
| RF7 | LS-SVR7 | Bi-LSTM7 | GA-BP7 | ✔ | ✔ | ✔ | ✔ | | | ✔ |

Figure 4 shows the flow chart of reference evapotranspiration model research. The data from 2000 to 2014 were taken as the training dataset, and the data from 2015 to 2020 were taken as the test dataset. Four different algorithms were combined to simulate $ET_0$ values in northern Xinjiang, and the simulation results were evaluated.

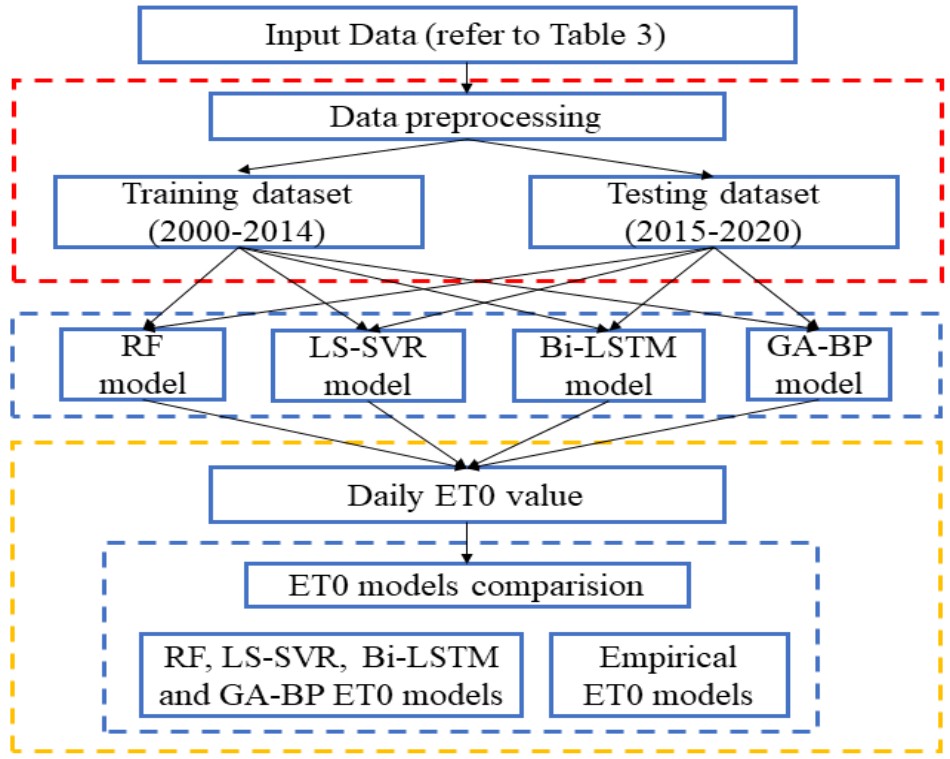

**Figure 4.** General representation of the $ET_0$ model study flowchart.

*3.7. Performance Evaluation of Models*

The assessment of the models was carried out using 5 statistical performance evaluations: the root mean square error (*RMSE*), the mean absolute error (*MAE*), the mean

bias error (*MBE*), correlation of determination ($R^2$) and global performance indicator (*GPI*) expressions, which are as follows:

$$RMSE = \sqrt{\frac{1}{n} \sum_{i=1}^{n} (ET_{P-M,i} - ET_{E,i})^2} \tag{18}$$

$$MAE = \frac{1}{n} \sum_{i=1}^{n} |ET_{P-M,i} - ET_{E,i}| \tag{19}$$

$$MBE = \frac{1}{n} \sum_{i=1}^{n} (ET_{P-M,i} - ET_{E,i}) \tag{20}$$

$$R^2 = 1 - \frac{\sum_{i=1}^{n} (ET_{P-M,i} - ET_{E,i})^2}{\sum_{i=1}^{n} (ET_{P-M,i} - \overline{ET_{P-M,i}})^2} \tag{21}$$

$$GPI = \sum_{i=1}^{4} \alpha_i (A_{i,new} - \overline{A_i}) \tag{22}$$

$$A_{i,new} = \frac{A_{i,old} - A_{i,min}}{A_{i,max} - A_{i,min}} \tag{23}$$

where $ET_{P\text{-}M,\,i}$ is the FAO56 P-M model-observed reference evapotranspiration at the *i*th time step (mm day$^{-1}$), $ET_{E,i}$ is the empirical or algorithm model (RF, LS-SV, Bi-LSTM and ANN models)-observed reference evapotranspiration at the *i*th time step (mm day$^{-1}$), and $n$ is the total observations. $A_{i,new}$ is the normalized value of the above four indicators, $\overline{A_i}$ corresponds to the median of the indicators, and $\alpha_i$ is equal to 1 for $R^2$, equal to $-1$ for the RMSE and MAE and equal to $\pm 1$ for the MBE indicators (the sign is the same as MBE).

## 4. Results and Analysis

### 4.1. Performance Appraisal of Seven Empirical Models (Temperature-Based and Mass Transfer-Based) for Estimating $ET_0$

Figure 5 shows that the three temperature-based empirical models have good applicability in northern Xinjiang, China, with $R^2$ values above 0.9. The Dorji model has the highest coefficient of determination, with $R^2$ up to 0.9275. However, the empirical model based on air mass transport is not ideal in northern Xinjiang, with an overall $R^2$ lower than 0.8, and the $R^2$ of the Albrecht model is only 0.7477. Compared with the 1:1 line, the Hargreaves model overestimate $ET_0$. The Berti model, Dorji model, Dalton model, Meyer model, WMO model and Albrecht model underestimated in northern Xinjiang. The average daily $ET_0$ of the Meyer model, WMO model and Albrecht model are all less than 2 mm, which is only 1/4 of that calculated by the P-M model, with a large error. Under the condition of wind speed $U_2$ and saturated vapour pressure difference ($e_s - e_a$) as the only inputs, it is impossible to accurately estimate $ET_0$ in this region. Among the seven empirical models, the Berti model and Hargreaves model have the best calculation effect in the northern region of Xinjiang. However, the WMO model and Meyer model have poor simulation effects. This finding indicates that the model based on air mass transport has poor applicability in northern Xinjiang, which further proves that temperature is the decisive factor affecting $ET_0$ in northern Xinjiang.

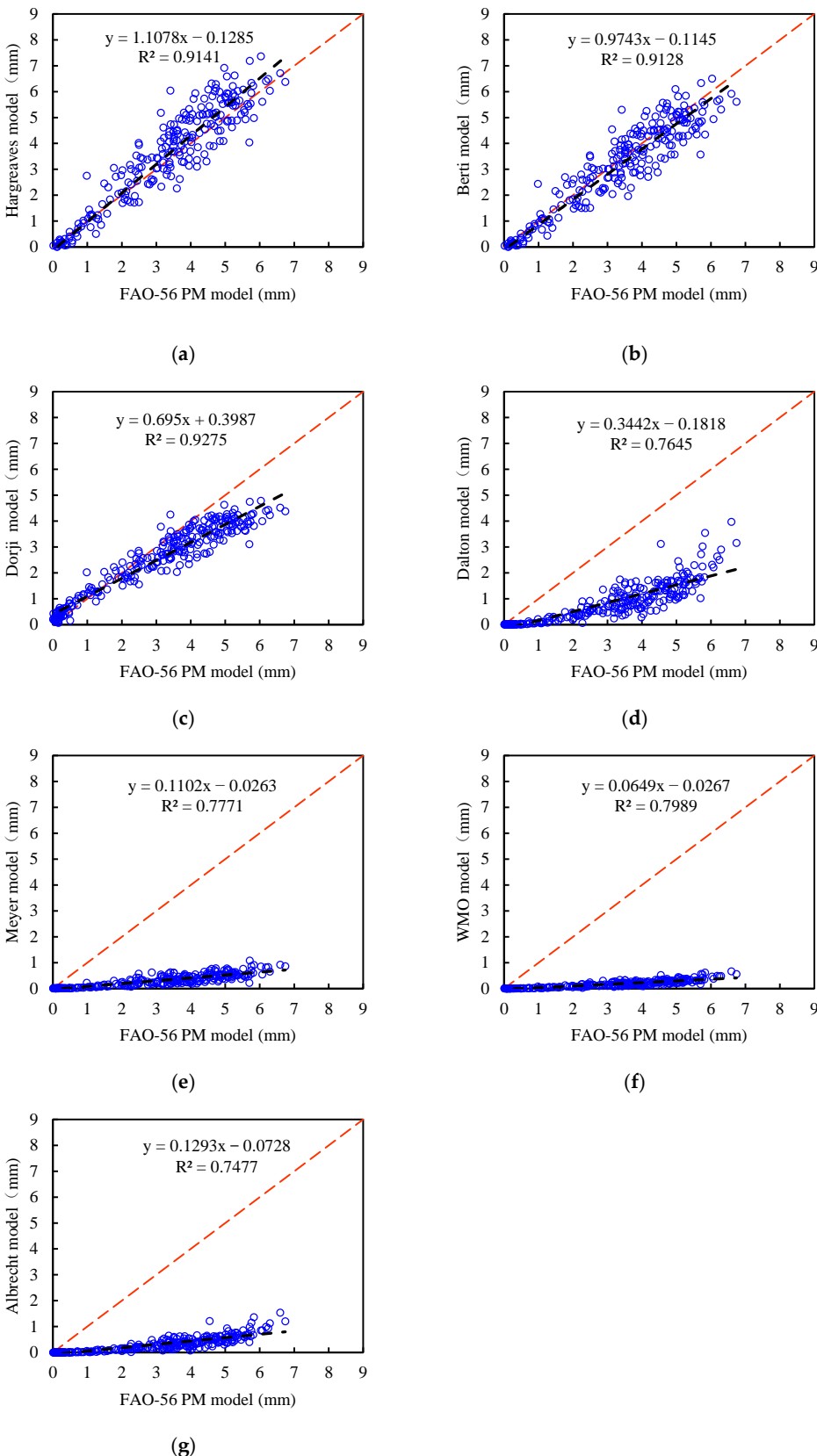

**Figure 5.** Comparisons of the daily reference evapotranspiration ($ET_0$) values calculated by the FAO-56 PM method vs. modelled data via the seven empirical models, including the Hargreaves model (**a**), Berti model (**b**), Dorji model (**c**), Dalton model (**d**), Meyer model (**e**), WMO model (**f**) and Albrecht model (**g**).

*4.2. Components Comparison of the Four Algorithm Models for Estimating ET$_0$*

In this study, seven different parameter combination input modes (Table 3) and four optimization algorithms, including the RF, LS-SVR, Bi LSTM and GA-BP models, were used to rank and evaluate the models in the training period (Table 4) and test period (Table 5) by introducing the global performance index (*GPI*). In the training period, the top 10% of the models were focused on the RF model, while the advantage of the RF model disappeared in the test period, which indicates that the RF model has overfitting in the training period, resulting in a significant reduction in the accuracy of the model in the test period. Therefore, the model evaluation in this paper is mainly based on the test period to reduce the impact of the overfitting algorithm on the simulation results.

Table 5 shows that the GA-BP5 model has the best overall simulation effect during the test period, with an *RMSE* of 0.2542 mm·d$^{-1}$, *MAE* of 0.1706 mm·d$^{-1}$, *MBE* of $-0.0039$ and $R^2$ of 0.9918. The second best simulation is the LS-SVR6 model, wherein the *RMSE* is 0.2380 mm·d$^{-1}$, *MAE* is 0.1604 mm·d$^{-1}$, *MBE* is $-0.0425$ and $R^2$ is 0.9928.

**Table 4.** Performance of RF, LS-SV, Bi-LSTM and GA-BP models during the training period.

| Models | Training Period (2000–2014) | | | | | |
|---|---|---|---|---|---|---|
| | *RMSE* (mm·d$^{-1}$) | *MAE* (mm·d$^{-1}$) | *MBE* | $R^2$ | *GPI* | Rank |
| RF1 | 0.2664 | 0.1672 | $-0.0006$ | 0.9891 | 0.4857 | 13 |
| RF2 | 0.2391 | 0.1493 | 0.0012 | 0.9912 | 0.7515 | 7 |
| RF3 | 0.2370 | 0.1487 | 0.0005 | 0.9914 | 0.7162 | 8 |
| RF4 | 0.2244 | 0.1385 | 0.0008 | 0.9923 | 0.8095 | 5 |
| RF5 | 0.1349 | 0.0873 | 0.0008 | 0.9972 | 1.2466 | 2 |
| RF6 | 0.1154 | 0.0075 | $-0.0004$ | 0.9980 | 1.4516 | 1 |
| RF7 | 0.1620 | 0.1032 | 0.0003 | 0.9960 | 1.0855 | 3 |
| LS-SVR1 | 0.5755 | 0.3659 | 0.0000 | 0.9491 | $-1.5234$ | 26 |
| LS-SVR2 | 0.5099 | 0.3267 | 0.0000 | 0.9601 | $-1.0476$ | 21 |
| LS-SVR3 | 0.5078 | 0.3246 | 0.0000 | 0.9604 | $-1.0304$ | 20 |
| LS-SVR4 | 0.4654 | 0.2975 | 0.0000 | 0.9667 | $-0.7329$ | 18 |
| LS-SVR5 | 0.2522 | 0.1665 | 0.0000 | 0.9902 | 0.5774 | 11 |
| LS-SVR6 | 0.2105 | 0.1447 | 0.0000 | 0.9932 | 0.7894 | 6 |
| LS-SVR7 | 0.3346 | 0.2154 | 0.0000 | 0.9828 | 0.1099 | 14 |
| BiLSTM1 | 0.5461 | 0.3541 | $-0.0006$ | 0.9542 | $-1.3624$ | 25 |
| BiLSTM2 | 0.5007 | 0.3226 | 0.0002 | 0.9615 | $-0.9742$ | 19 |
| BiLSTM3 | 0.5285 | 0.3373 | $-0.0004$ | 0.9571 | $-1.2048$ | 23 |
| BiLSTM4 | 0.4678 | 0.2972 | 0.0002 | 0.9664 | $-0.7319$ | 17 |
| BiLSTM5 | 0.2648 | 0.1785 | 0.0000 | 0.9892 | 0.4959 | 12 |
| BiLSTM6 | 0.2342 | 0.1587 | $-0.0006$ | 0.9916 | 0.6282 | 10 |
| BiLSTM7 | 0.3518 | 0.2310 | 0.0000 | 0.9810 | $-0.0084$ | 16 |
| GA-BP1 | 0.5658 | 0.3591 | $-0.0060$ | 0.9508 | $-1.8223$ | 28 |
| GA-BP2 | 0.5041 | 0.3295 | $-0.0006$ | 0.9610 | $-1.0606$ | 22 |
| GA-BP3 | 0.5071 | 0.3289 | $-0.0047$ | 0.9605 | $-1.3323$ | 24 |
| GA-BP4 | 0.4761 | 0.3109 | $-0.0139$ | 0.9652 | $-1.6939$ | 27 |
| GA-BP5 | 0.2508 | 0.1692 | 0.0021 | 0.9903 | 0.7063 | 9 |
| GA-BP6 | 0.2145 | 0.1504 | 0.0011 | 0.9929 | 0.8261 | 4 |
| GA-BP7 | 0.3256 | 0.2129 | $-0.0014$ | 0.9837 | 0.0659 | 15 |

The worst model is the Bi-LSTM1 model, wherein the *RMSE* is 0.7811 mm·d$^{-1}$, *MAE* is 0.5374 mm·d$^{-1}$, *MBE* is $-0.3776$ and $R^2$ is 0.9227. The worst Bi-LSTM1 model has the least number of input parameters, and the best, the GA-BP5 model and LS-SVR6 model, have the most input parameters, which shows that the number of input parameters has a great impact on the model simulation accuracy. In the same algorithm with different

parameter combinations, the best results of RF model and LS-SVR model appear in the sixth group of parameter inputs ($T$, $T_{max}$, $T_{min}$, $Ra$, $Rs$ and $U_2$), and the best results of Bi LSTM model and GA-BP model appear in the fifth group of parameter inputs ($T$, $T_{max}$, $T_{min}$, $Ra$, $Rs$ and $U_2$); this finding shows that the accuracy of the model cannot be improved by 100% only by increasing the number of input parameters.

**Table 5.** Performance of RF, LS-SV, Bi-LSTM and GA-BP models during the testing period.

| Models | Testing Period (2015–2020) | | | | | |
|---|---|---|---|---|---|---|
| | *RMSE* (mm·d$^{-1}$) | *MAE* (mm·d$^{-1}$) | *MBE* | $R^2$ | *GPI* | Rank |
| RF1 | 0.7805 | 0.5168 | −0.3149 | 0.9229 | −1.6354 | 27 |
| RF2 | 0.7161 | 0.4633 | −0.2893 | 0.9351 | −1.1373 | 18 |
| RF3 | 0.7159 | 0.4631 | −0.2903 | 0.9351 | −1.1386 | 19 |
| RF4 | 0.7003 | 0.4457 | −0.2979 | 0.9379 | −1.0424 | 17 |
| RF5 | 0.3492 | 0.2276 | −0.0440 | 0.9846 | 1.4763 | 9 |
| RF6 | 0.3201 | 0.2043 | −0.0720 | 0.9870 | 1.5575 | 7 |
| RF7 | 0.4156 | 0.2790 | −0.0675 | 0.9781 | 1.0678 | 11 |
| LS-SVR1 | 0.7522 | 0.4963 | −0.3260 | 0.9284 | −1.4778 | 24 |
| LS-SVR2 | 0.7146 | 0.4646 | −0.3314 | 0.9353 | −1.2383 | 20 |
| LS-SVR3 | 0.6593 | 0.4293 | −0.2770 | 0.9450 | −0.7710 | 14 |
| LS-SVR4 | 0.6366 | 0.4101 | −0.2882 | 0.9487 | −0.6528 | 13 |
| LS-SVR5 | 0.2621 | 0.1724 | −0.0062 | 0.9913 | 1.9727 | 4 |
| LS-SVR6 | 0.2380 | 0.1604 | −0.0425 | 0.9928 | 1.9809 | 2 |
| LS-SVR7 | 0.3548 | 0.2371 | −0.0455 | 0.9841 | 1.4300 | 10 |
| BiLSTM1 | 0.7811 | 0.5374 | −0.3776 | 0.9227 | −1.8483 | 28 |
| BiLSTM2 | 0.7400 | 0.4925 | −0.3797 | 0.9307 | −1.5453 | 26 |
| BiLSTM3 | 0.7189 | 0.4799 | −0.3198 | 0.9346 | −1.2692 | 22 |
| BiLSTM4 | 0.7011 | 0.4647 | −0.3570 | 0.9378 | −1.2427 | 21 |
| BiLSTM5 | 0.2787 | 0.1831 | 0.0245 | 0.9902 | 1.9735 | 3 |
| BiLSTM6 | 0.3033 | 0.2022 | −0.0877 | 0.9884 | 1.5742 | 6 |
| BiLSTM7 | 0.4287 | 0.2990 | −0.0887 | 0.9767 | 0.9184 | 12 |
| GA-BP1 | 0.7572 | 0.5009 | −0.3312 | 0.9274 | −1.5261 | 25 |
| GA-BP2 | 0.7330 | 0.4832 | −0.3578 | 0.9320 | −1.4350 | 23 |
| GA-BP3 | 0.6673 | 0.4405 | −0.2923 | 0.9436 | −0.8724 | 16 |
| GA-BP4 | 0.6571 | 0.4314 | −0.3058 | 0.9453 | −0.8388 | 15 |
| GA-BP5 | 0.2542 | 0.1706 | 0.0039 | 0.9918 | 2.0245 | 1 |
| GA-BP6 | 0.2434 | 0.1684 | −0.0480 | 0.9925 | 1.9312 | 5 |
| GA-BP7 | 0.3407 | 0.2273 | −0.0331 | 0.9853 | 1.5301 | 8 |

Because the influence of meteorological factors on reference evapotranspiration is different, considering $RH$ and $U_2$, $RMSE$ decreases by 0.5 mm·d$^{-1}$, $MAE$ decreases by 0.32 mm·d$^{-1}$, $MBE$ amplitude decreases by 0.3 and $R^2$ increases by 0.077. Considering the radiation terms $Rs$ and $U_2$ based on the temperature model, $RMSE$ decreases by 0.52 mm·d$^{-1}$, $MAE$ decreases by 0.33 mm·d$^{-1}$, $MBE$ amplitude decreases by 0.28 and $R^2$ increases by 0.076. The results show that the radiation term $Rs$ and aerodynamic term $U_2$ have a significant effect on reference evapotranspiration, and the reference evapotranspiration model is mainly composed of three parts: the temperature term, radiation term and aerodynamic term.

We obtained the optimal model results under different input combinations (Table 6). Figure 6 shows a comparison between the optimal model and the P-M model under the conditions of various input parameters, which can also support the view that the accuracy of simulation results is improved after considering the radiation term $Rs$ and aerodynamic term $U_2$ parameters.

**Table 6.** The optimal model of different parameter combination input.

| Optimal Model | $T$ | $T_{max}$ | $T_{min}$ | $Ra$ | $Rs$ | $RH$ | $U_2$ |
|---|---|---|---|---|---|---|---|
| LS-SVR1 | ✔ | ✔ | ✔ | ✔ | | | |
| RF2 | ✔ | ✔ | ✔ | ✔ | ✔ | | |
| LS-SVR3 | ✔ | ✔ | ✔ | ✔ | | ✔ | |
| LS-SVR4 | ✔ | ✔ | ✔ | ✔ | ✔ | ✔ | |
| GA-BP5 | ✔ | ✔ | ✔ | ✔ | | ✔ | ✔ |
| LS-SVR6 | ✔ | ✔ | ✔ | ✔ | ✔ | | ✔ |
| GA-BP7 | ✔ | ✔ | ✔ | ✔ | | | ✔ |

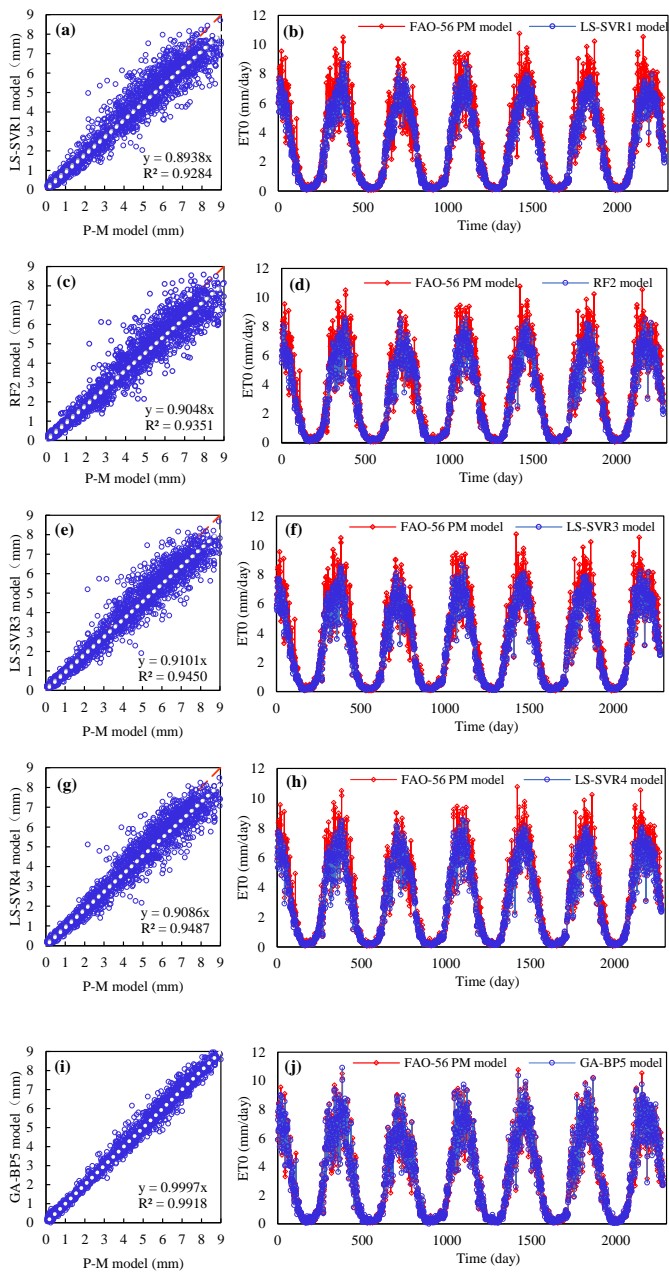

**Figure 6.** *Cont.*

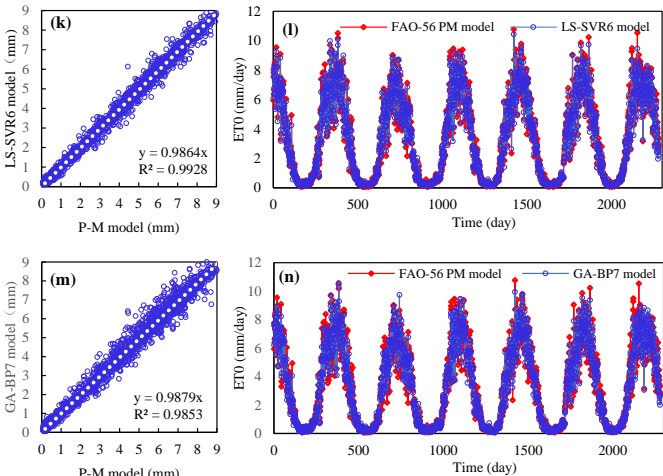

**Figure 6.** Comparisons of the daily reference evapotranspiration ($ET_0$) values calculated by the FAO-56 PM method vs. modelled data via the optimal reference evapotranspiration model under different input combinations during the testing phase. LS-SVR1 model correlation diagram (**a**), LS-SVR1 model measured value and simulated value (**b**); RF2 model correlation diagram (**c**), RF2 model measured value and simulated value (**d**); LS-SVR3 model correlation diagram (**e**), LS-SVR3 model measured value and simulated value (**f**); LS-SVR4 model correlation diagram (**g**), LS-SVR4 model measured value and simulated value (**h**); GA-BP5 model correlation diagram (**i**), GA-BP5 model measured value and simulated value (**j**); LS-SVR6 model correlation diagram (**k**), LS-SVR6 model measured value and simulated value (**l**); GA-BP7 model correlation diagram (**m**), GA-BP7 model measured value and simulated value (**n**).

## 4.3. Evaluation of Optimal Reference Evapotranspiration Model under Different Time-Scale Conditions

In the verification of the empirical model and optimization algorithm model, this paper takes the data from 2000 to 2020 as samples. A large amount of data and a long time span increase the model verification accuracy. Generally, the estimation of crop water requirements is only based on the reference crop evapotranspiration of 5–7 days. Therefore, the influence of different time scales on the simulation accuracy of the model is also included in the scope of this paper. We divided our data into four time scales, which were 7 days (1 July 2015 to 7 July 2015), one month (May 2019), one season (summer 2015) and one year (2017), and evaluated them with the Berti model, Hargreaves model and optimal model under seven different parameter input combinations. Figure 7 shows a comparison of Taylor diagrams of models under different time scales. It can be seen from Figure 7, when the time scale is 7 days, the range of standard deviation is 0.5~1.25, while, when the time scale is year, the range of standard deviation is reduced to 0.625~1, and the correlation is increased from 0.75 to 0.95. This result shows that the time scale has a significant effect on the accuracy of the model, and the accuracy of the model increases with increasing time scale. At the annual scale, the seven models based on the optimization algorithm are better. When meteorological data are missing, seven models can be used to estimate $ET_0$ in northern Xinjiang. Only the LS-SVR6 model and GA-BP7 model are recommended to estimate the local $ET_0$ at the daily, monthly and seasonal scales.

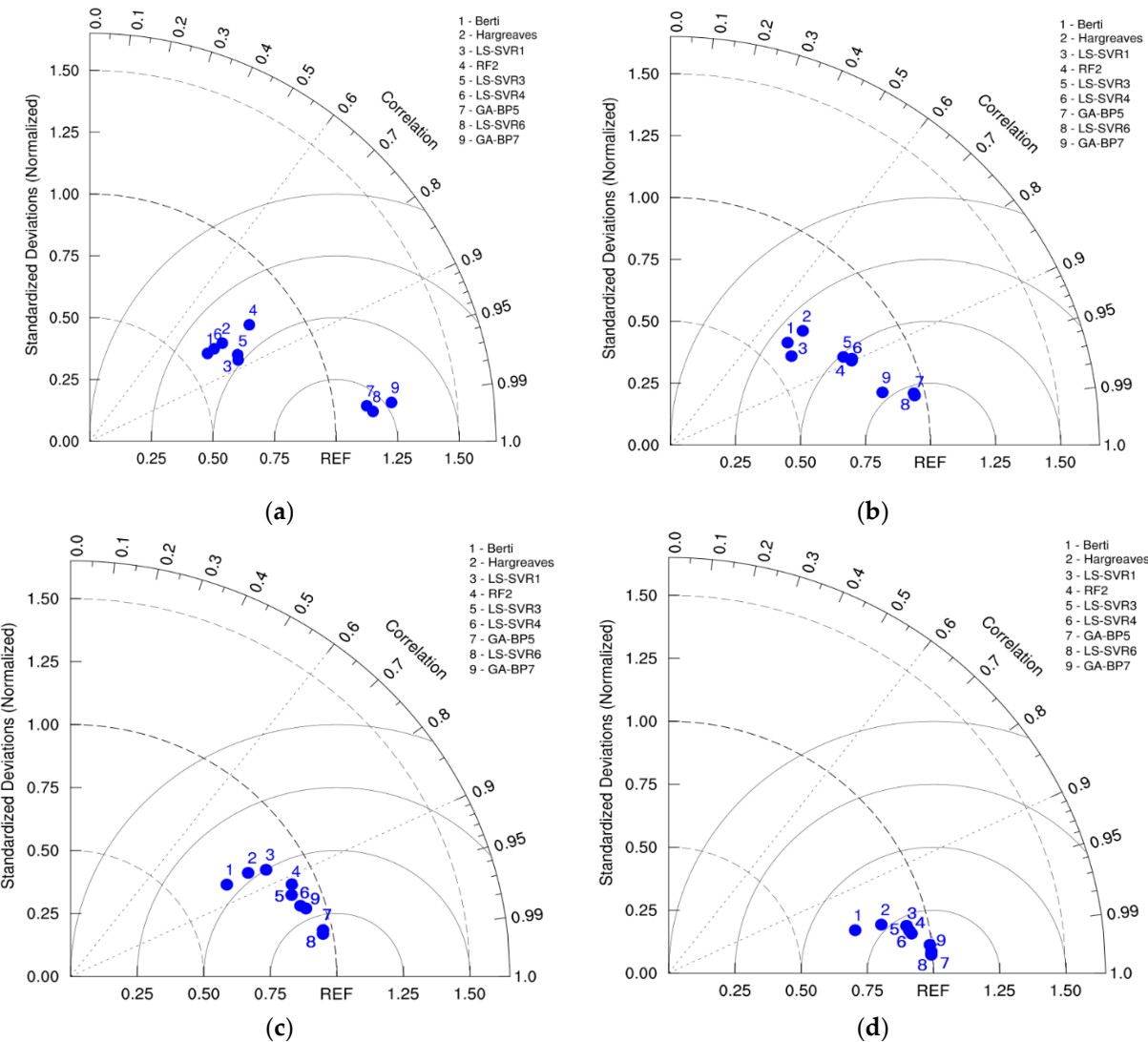

**Figure 7.** Evaluation of Taylor diagram for reference evapotranspiration model in different time scales; 7 days (**a**), May 2019 (**b**), quarter 2015 (**c**), 2016 (**d**).

## 5. Discussion

The correlation between reference evapotranspiration model and meteorological factors is the key to determining the input parameters of the optimization model, and many studies have shown that air temperature is the primary factor affecting reference evapotranspiration [4,5,7]. This study shows that $ET_0$ has a very significant positive correlation with the daily average temperature $T$, the daily minimum air temperature $T_{min}$, the daily maximum air temperature $T_{max}$, the net radiation Ra and the saturated water vapor pressure difference *VPD* at the level of $p < 0.001$ (Figure 3), and the saturated water air pressure difference *VPD* is a function of temperature and humidity. The relative humidity *RH* was negatively correlated with $ET_0$ at the level of $p < 0.001$. The whole evapotranspiration process is dominated by temperature and net radiation *Ra*. This result is consistent with the conclusions of some scholars [22–24].

This study found that among the empirical models, the temperature-based model is more suitable for northern Xinjiang, this is consistent with the conclusion of Rachid et al. [34]. That is, the temperature-based model is more suitable for arid and semi-arid areas. The reason is that heat is the main driving factor of water transmission and the key step of the water cycle, especially in arid and semi-arid areas with limited water resources. It is feasible and reliable to apply the optimization algorithm to $ET_0$ prediction. However,

under the same parameter input conditions, the estimation accuracy of $ET_0$ at different scales is different, this conclusion is consistent with the views of some scholars [35–37]. The main reason is that the time scale itself is the factor restricting the accuracy of the algorithm. It can be seen from Figure 7 that the model based on the optimization algorithm is significantly better than the empirical, and this conclusion has also been verified in semi-arid areas of Spain [38]. Chen et al. [12] also pointed out that the model based on optimization algorithm is better than the empirical model, and the empirical model based on temperature factor is better than that based on radiation or humidity in the training and research area. It shows that the application of an optimization algorithm can greatly improve the estimation accuracy of the model and make up for the shortcomings of existing empirical formulas.

## 6. Conclusions

Based on the daily datasets of eight meteorological stations in northern Xinjiang, China, from 2000 to 2020, seven empirical models (Hargreaves, Berti, Dorji, Dalton, Meyer, WMO, and Albrecht) and 28 models based on optimization algorithms (RF, LS-SVR, Bi-LSTM and GA-BP) were compared. Through the evaluation of eight empirical models, the Berti model and Hargreaves model are better alternative models, while the WMO model and Meyer model are worse alternative models. The global performance index (GPI) is used to rank 28 models based on the optimization algorithm, and the optimal recommendation models of the annual scale and multi-year scale under different input parameter combinations are obtained as follows: LS-SVR1 (input: $T$, $T_{max}$, $T_{min}$, $Ra$), RF2 (input: $T$, $T_{max}$, $T_{min}$, $Ra$, $Rs$), LS-SVR3 (input: $T$, $T_{max}$, $T_{min}$, $Ra$, $RH$), LS-SVR4 (input: $T$, $T_{max}$, $T_{min}$, $Ra$, $Rs$, $RH$), GA-BP5 (input: $T$, $T_{max}$, $T_{min}$, $Ra$, $RH$, $U_2$), LS-SVR6 (input: $T$, $T_{max}$, $T_{min}$, $Ra$, $Rs$, $U_2$) and GA-BP7 (input: $T$, $T_{max}$, $T_{min}$, $Ra$, $U_2$). Only the LS-SVR6 model and GA-BP7 model are recommended as the best alternative models for $ET_0$ when the local climate dataset is incomplete at the daily, monthly and seasonal scales, which has high applicability in northern Xinjiang. This study can provide theoretical guidance and technical support for the determination of farmland irrigation systems and water resource planning and management in northern Xinjiang. This method is also applicable to other arid and semi-arid areas.

**Author Contributions:** Conceptualization, P.J. and S.-J.H.; methodology, P.J.; software, P.J.; validation, P.J.; formal analysis, P.J.; investigation, P.J. and S.-J.H.; resources, S.-J.H.; data curation, P.J. and S.-J.H.; writing—original draft preparation, P.J.; writing—review and editing, P.J. and S.-J.H.; supervision, S.-J.H. All authors have read and agreed to the published version of the manuscript.

**Funding:** This work was supported by the special fund project of Xinjiang Water Conservancy Science and technology (No. YF2020-08 and XSK-2021-05), the National Natural Science Foundation of China (No. 41671032), the National Key R&D Program of China (No. 2013CB429,902).

**Institutional Review Board Statement:** Not applicable.

**Informed Consent Statement:** Not applicable.

**Data Availability Statement:** Data is contained within article.

**Conflicts of Interest:** The authors declare no conflict of interest.

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
