# Peer review of "Optimal Alternative for Quantifying Reference Evapotranspiration in Northern Xinjiang"

_water, doi:10.3390/w14010001_

Round 1
Reviewer 1 Report
The reviewer comments on the attachment

Author Response
Point 1: The Penman-Monteith method (PM) is recommended as the sole standard method for estimating reference evapotranspiration (ET0). The main limitation of this method is the difficulty in obtaining all necessary input data (air temperature, humidity, solar radiation, and wind speed). In such circumstances, simple equations are often used to estimate ET0. why these alternative models are optimal?
Response 1: When the obtained meteorological factors are limited, the P-M model cannot be used to calculate the evapotranspiration. At this time, the evapotranspiration can be estimated through the limited meteorological factors through the empirical model and the model based on the optimization algorithm. This study mainly judges the advantages and disadvantages of the model through the evaluation indexes (RMSE, MAE, MBE, R2, GPI).
Point 2: References are not always relevant and up-to-date. This reviewer suggests exclusion of few publications (30, 33, 37).
Response 2: Deleted as required
Point 3: Mention the fact that many drought indices (RDI, SPEI, WSVI) are based on evapotranspiration.
Response 3: Supplemented, please review
Point 4: What are the tuning parameters of RF, LSTM, SVM? Are data quality control?
Response 4: Yes. A total of 8 sites were selected in this study, of which, the Wujiaqu, Wusu, Shawan and Qinghe datasets span from 2000 to 2017, and the datasets of Hutubi, Fuyun, Hebukesel and Karamay span from 2000 to 2020. The site with missing data was cancelled in 2018. Data quality control is mainly eliminated by screening outliers.
Point 5: Scholarly discussion is missing in the paper. This paper lacks of comparison with other studies.
Response 5: Supplemented, please review

Reviewer 2 Report
The manuscript water-1501662, entitled “Optimal alternative for quantifying reference evapotranspiration in Northern Xinjiang” submitted by Ping Jiao and Jun Shun Hu report the results related to a work of ET0 calculation and validation that involved 28 optimization algorithm derived from the combination of 8 empirical models and 4 optimization algorithms. In order to evaluate and validate the results achieved along the 2000-2020 period, the models were compared with the standard FAO PM equation. The climatic data used were obtained from 8 different stations located in contrasting environments within Northern Xinjiang region.
Considering the importance of an accurate ET0 calculation, in order to improve water management efficiency and avoid crop stress, also in the context where the climatic information is limited, I believe that the manuscript is of potential interest for readers of “Water” and fall within its scope.
In general, the experimental activity was carried out following a strict scientific logic and according to widely used methods which have made it possible to obtain reliable results. Indeed, the authors applied an appropriate calculation plan design, analyses follow a solid logic allowing to achieve reliable results.
The quality of the writing is good and only small changes are needed.
The introduction is fine and adequate and only small changes are needed. However, it is quite long. If possible, I suggest condensing some parts.
The results are clear and easy to read. The image quality is adequate.
Discussion is absent and paragraph 5 represent a continuation of the results. In the discussion, the authors should try to argue and comment on the results obtained by comparing them with the results observed by other authors, and finally, give advice to the end-users of the methods they suggest.
Conclusions are fine.
My specific comments, which I hope will help the authors to improve the manuscript, are reported in the attached files.

Author Response
Point 1: not is the only equation can be used but is the best for that. change this word or just delete "only"
Response 1: Deleted as required
Point 2: (Performance of 28 optimization algorithm models formed by input combinations (T, Tmax, Tmin, Ra, Rs, RH, and U2).) this second part is not well connected to the first period. i suggest rearranging the sentence or deleting it.
Response 2: Deleted as required
Point 3: Delete 31 line "only"
Response 3: Deleted as required
Point 4: “this method” change this” its application”
Response 4: Modified, please review
Point 5: (combined model is the best) convert to plural
Response 5: Modified, please review
Point 6: (Combined with the parameters measured by China's meteorological stations, there are a lack of radiation parameters, and surface radiation parameters need to be downloaded separately) it is difficult to understand. please, rewrite.
Response 6: Modified, please review
Point 7: (However, the combined model is highly dependent on the input of meteorological variables, which requires the sufficient input of meteorological variables) rewrite in order to avoid repetitions.
Response 7: Modified, please review
Point 8: water and gas deficiency
Response 8: Modified, please review (Water vapor pressure difference)
Point 9: (Therefore, on the basis of the FAO56-PM model, it is crucial to verify an alternative model with limited meteorological variables, simple calculations and good simulation effects.) it seems a partial repetition of the period before. Please, reorganize/join the two paragraph in order to improve readability.
Response 9: Modified, please review
Point 10: (which is the novelty of this study.) delete
Response 10: Deleted as required
Point 11: (To date, in the case of limited meteorological data, the performance and application of the evapotranspiration model in northern Xinjiang still need to be further verified.) delete
Response 11: Deleted as required
Point 12: move the text above figure 4, 6
Response 12: Modified, please review
Point 13: (This is not a discussion but a continuation of the results. Discussion of results is absent and needs to be implemented.) rewrite
Response 13: Rewritten, please review

Reviewer 3 Report
The article contains interesting analyses on the estimation of reference evapotranspiration. The authors have checked the models in detail and performed the statistical analysis correctly. I am impressed with this manuscript. However, it has one disadvantage. The discussion section is poorly written. There is a lack of references to similar analyses - attempts to analyse results from individual models. Please supplement this with worldwide references. Moreover, I have some minor editing comments: In the title, every word should be capitalized. There is an error in the abstract. The word,,Performance" (line 18) should be lower case or there should be a full stop before it. In line 383 pay attention to the notation of units. It should be multiplication i.e. mm*d-1, not division. Same in table 4. Please replace mm/d with the correct notation. In line 110 you use abbreviations T, Tmax, Tmin, Ra, Rs, RH and U2 which are not explained earlier. Please first explain what these data are and then use the abbreviation of their names. In Table 1 you also for the first time use abbreviations that you have not explained before - please change this. In figure 1 you have the description elevation (m).tif please remove the word ,,tif" - it is not necessary.Author Response
The article contains interesting analyses on the estimation of reference evapotranspiration. The authors have checked the models in detail and performed the statistical analysis correctly. I am impressed with this manuscript.
Point 1: However, it has one disadvantage. The discussion section is poorly written. There is a lack of references to similar analyses - attempts to analyse results from individual models. Please supplement this with worldwide references.
Response 1: Modified, please review
Point 2: Moreover, I have some minor editing comments: In the title, every word should be capitalized.
Response 2: Modified, please review
Point 3: There is an error in the abstract. The word,,Performance" (line 18) should be lower case or there should be a full stop before it.
Response 3: Modified, please review
Point 4: In line 383 pay attention to the notation of units. It should be multiplication i.e. mm*d-1, not division. Same in table 4. Please replace mm/d with the correct notation.
Response 4: Modified, please review
Point 5: In line 110 you use abbreviations T, Tmax, Tmin, Ra, Rs, RH and U2 which are not explained earlier. Please first explain what these data are and then use the abbreviation of their names.
Response 5: Modified or deleted, please review
Point 6: In Table 1 you also for the first time use abbreviations that you have not explained before - please change this.
Response 6: Modified, please review
Point 7: In figure 1 you have the description elevation (m).tif please remove the word, tif" - it is not necessary.
Response 7: Deleted, please review

Round 2
Reviewer 1 Report
Authors have addressed all the raised concern. Now, it can be accepted for publication
Reviewer 2 Report
The authors applied all the requested suggestions and also implemented the discussion part. I am totally satisfied with the implemented changes.